# Pilot study to establish a prospective neonatal cohort: Study of Preterm Infants and Neurodevelopmental Genes (SPRING)

Hilary S Wong [iD],[1] Lucinda Hopkins,[2] Michael C O'Donovan,[2] Anita Thapar,[2] Neena Modi[3]

► Additional material is published online only. To view, please visit the journal online (http://dx.doi.org/10.1136/bmjpo-2020-000648).

[1]Department of Paediatrics, University of Cambridge School, Cambridge, UK
[2]MRC Centre for Neuropsychiatric Genetics and Genomics, Cardiff University, Cardiff, UK
[3]Section of Neonatal Medicine, Imperial College London Department of Medicine, London, UK

**Correspondence to**
Professor Neena Modi; n.modi@imperial.ac.uk

## ABSTRACT

**Background** Genetic risk variants and preterm birth are early and potent risk factors for later neuropsychiatric disorders. To understand the interrelationships between these factors, a large-scale genetic study of very preterm (VPT, <32 weeks gestation) infants with prospective follow-up is required. In this paper, we describe a streamlined study approach, using efficient processes for biological and clinical data collection, to feasibly establish such a cohort.

**Methods** We sought to recruit 500 VPT families within a 1 year period from neonatal units. Treating clinical teams recruited eligible participants, obtained parent consent, collected blood samples and posted specimens to the research laboratory. We extracted all clinical data from the National Neonatal Research Database, an existing UK resource that captures daily patient-level data on all VPT infants.

**Results** Between May 2017 and June 2018, we established a cohort of 848 VPT infants and their parents from 60 English neonatal units. The study population (median (IQR), gestation: 28.9 (26–30) weeks; birth weight: 1120 (886–1420) g) represented 18.9% of eligible infants born at the study sites during the recruitment period (n=4491). From the subset of 521 complete family trios, we successfully completed genotyping for 510 (97.9%) trios. Of the original 883 infants whose parents consented to participate, the parents of 796 (90.1%) infants agreed to future data linkage and 794 (89.9%) agreed to be recalled.

**Conclusion** We demonstrate the feasibility and acceptability of streamlined strategies for genetic, neonatal and longitudinal data collection and provide a template for future cost-effective and efficient cohort development.

## BACKGROUND

Globally, approximately 15 million infants are born preterm (<37 weeks gestation) each year and the prevalence is rising.[1] The causes of spontaneous preterm births and the conditions justifying 'medically indicated' preterm birth are complex and poorly understood, with both environmental factors[2 3] and genetic components[4] implicated. Preterm infants experience significant adversities and are at elevated risk of intellectual disability,

### What is known about the subject?

► Infants born very preterm are at increased risk of developing neuropsychiatric conditions such as attention-deficit hyperactivity disorder and autism spectrum disorder.
► A large number of genetic risk variants have been identified for neuropsychiatric disorders.

### What this study adds?

► A large-scale efficient and cost-effective cohort study of preterm infants for genetic investigations in the UK is feasible.
► Longitudinal health and education outcome ascertainment of very preterm infants through routine data linkage are acceptable to parents.

behavioural and mental health problems, with approximately a quarter of children born extremely preterm meeting neuropsychiatric diagnostic criteria by childhood.[5] However, there is considerable heterogeneity in outcomes, the causes of which are unknown.

Previous epidemiological studies have suggested family and genetic factors may contribute to the link between preterm birth and later neuropsychiatric disorders. Many of these studies have observed an association between maternal mental health problems and risk of preterm birth.[6 7] Positive family history of psychiatric disorder further contributes to the risk of adverse mental health outcomes among individuals born preterm.[8 9] Genomic studies have identified large numbers of genetic risk variants for major neuropsychiatric disorders, including those for attention-deficit hyperactivity disorder and autism spectrum disorder, which are the neuropsychiatric disorders most

strongly associated with preterm birth in childhood.[10] However, the relevance of these genetic risk variants for those born preterm has not been investigated.

A large preterm birth cohort with genomic data and prospective clinical phenotyping would be required to investigate the relationship between genetic risks, preterm birth and neuropsychiatric outcomes. There is no sufficiently powered existing prospective cohort of preterm infants for such investigations and the costs, time and resource requirements for recruitment and data collection by established methods are likely to be prohibitive. Our aims were to examine the feasibility of large-scale recruitment and DNA collection in a multicentre setting, by capitalising on existing research infrastructure in the UK to streamline operational efficiencies, and the acceptability to parents of prospective follow-up through routine data linkage and direct recall for detailed phenotyping. Our secondary aim was to conduct a preliminary investigation to explore the hypothesis that the very preterm (VPT, <32 weeks gestation) population is enriched for rare pathogenic copy number variants; the results of this investigation were reported in a separate paper.[11]

## METHODS
### Study sites and recruitment
We invited expressions of interest in participation as study sites from all 43 neonatal intensive care units (NICUs) and 77 local neonatal units (LNUs) in England. In the UK, neonatal care is provided within managed clinical networks where LNUs provide the full range of care, including short periods of intensive care, for babies delivered in the nominal catchment area of the hospital.[12] Most babies born over 27 weeks gestation will usually receive their full care in their LNU. Babies who are more preterm or those who require complex or longer-term intensive care are transferred to a NICU, where tertiary specialist care is provided. From the units that expressed interest, we selected study sites based on the potential number of eligible participants, geographical coverage and record of accomplishment of research activities in the unit. The study was included in the National Institute for Health Research Clinical Research Network Portfolio, which allowed us to access support, such as Clinical Research Network-funded research nurses in England. Being a 'portfolio study' also acts as an incentive for recruitment activities from neonatal units as the accrual data are used to inform future allocations of NHS research support funding to hospital trusts. All study procedures were carried out by the treating clinical or research staff based locally at the participating study sites. These local staff received training in the study procedures from the investigator team using training videos and teleconferences.

Recruitment took place between 1 May 2017 and 30 June 2018. Eligible participants were infants born VPT and their parents. While we aimed to recruit both parents

for trio-based genome analysis, single-parent families were also included. Similarly, we recruited infants where one or both parents declined to participate in the study themselves. We did not recruit parents known to be genetically unrelated to the child, for instance if the pregnancy had arisen from a donor gamete. Potential participants could be approached and recruited at any point during their neonatal hospitalisation at the discretion of the local research staff. Informed written consent was obtained from each parent (for themselves) and their offspring at the local site. We sought to recruit 500 VPT infants over the 1 year recruitment phase.

### Biological specimen collection and laboratory processing
Blood samples from participants were collected into EDTA specimen bottles. For infants, 0.5 to 1 mL of blood were collected by heel lance, venepuncture or from an indwelling arterial catheter during blood sampling for routine clinical care to avoid an additional intervention for the infant. If the target volume was not achieved, a second aliquot was obtained, also collected during routine sampling. Five millilitre of blood was collected from each parent. The blood specimens were labelled using barcodes and a unique participant study number, packaged in a designated mailing box for biological specimens, in accordance with UN3373 regulations and P650 packaging instructions, and posted to the research laboratory at the MRC Centre for Neuropsychiatric Genetics and Genomics (MRC CNGG), Cardiff University within 7 days of sample collection using UK standard postal services. No laboratory processing of the samples took place at the recruitment study sites.

DNA was extracted from the blood specimens in-house at the MRC CNGG, from full sets of parent–infant trios in the first instance, using GE Healthcare DNA extraction kits. DNA sample quantification was determined using Quant-iT PicoGreen dsDNA assay kits and samples were genotyped using Illumina OmniExpress BeadChip arrays. Samples with concentrations below 50 ng/μL were excluded from genotyping. DNA was considered of adequate quality on successful completion of array-based genotyping.

### Clinical data collection
To minimise research workload and participant burden, we capitalised on the existing National Neonatal Research Database (NNRD) to access relevant clinical data on the research participants. Investigator NM led the establishment of the NNRD in 2007 to support the use of clinical data to improve neonatal care and outcomes, and the UK Neonatal Collaborative comprising all neonatal units that contributed data. Patient-level point-of-care electronic data entered by treating clinicians on all infants admitted to the neonatal units are released with permission from the Caldicott Guardians to the Neonatal Data Analysis Unit at Imperial College London, where they are merged and cleaned to create the NNRD (https://www.imperial.ac.uk/neonatal-data-analysis-unit). Data captured on the

NNRD include demographic details, daily records of intervention and treatments through the neonatal inpatient stay, diagnoses and follow-up health status at age 2 years. The NNRD has complete population coverage of all neonatal unit admissions, including all VPT births in England, Wales and Scotland (approximately 7000 annually) and is approved by the UK National Research Ethics Service (10/80803/151) as a research database and by the Health Research Authority Confidentiality Advisory Group (8-05(f)/0210).

On receipt of parent signed authorisation, the unique NNRD identifier for each participating infant was obtained from his/her neonatologist. All relevant participant clinical data, including gestational age, birth weight, sex, time of onset of labour and maternal postcode, were extracted from the NNRD.

### Consent for follow-up

Permissions were sought from parents of participating infants to conduct follow-up studies by means of linkage of the genetic and clinical data to future routine health and educational records, in order to reduce the research burden associated with long-term follow-up and outcome ascertainment. This was indicated by the parents' response to the following statement in the consent form: '*I give permission for the research team to link the study information on my baby to records that the NHS and other public organisations hold*'. Consent was also sought for recontacting the family in the future for detailed phenotyping through face-to-face or questionnaire-based assessments. Parents were made aware of their right to withdraw from the study at any point by contacting the researchers.

### Patient and public involvement

We involved Bliss, the leading national preterm charity in the design and funding application for the study. We consulted parent representatives in developing the study materials including the participant information sheet. They will be consulted for the public dissemination of the findings from this research.

### RESULTS
### Recruitment and establishment of study population

We received expressions of interest from 75 out of 120 neonatal units in England. Of these, we selected 60 neonatal units (24 NICUs and 36 LNUs) to be study sites. These provided an extensive population catchment coverage in England (see online supplementary material 1 for geographical coverage and list of participating hospitals).

During the 14-month recruitment period, 1853 families were approached and 779 (42.0%) families, consisting of 883 infants, consented to participation (figure 1). Blood samples were not collected in nine infants prior to their discharge. In five cases, the sample was lost during postage. Blood samples from 869 infants, 753 mothers and 551 fathers were successfully received in the MRC CNGG laboratory. Of these samples, there were 529 full infant–parent trios, quads (in twin births) or quints (in triplet births).

Clinical data extraction from the NNRD was successful in 848 (97.6%) infants whose blood samples had arrived at the laboratory. The demographic and neonatal characteristics of these participants are summarised in table 1.

During the recruitment period, a total of 8998 infants were born at <32 weeks gestation in England, with 4491 being born at participating study sites. Therefore, this study cohort represent 18.9% of eligible infants born at study sites and 9.4% of the national VPT population. Our study sample was representative of the national VPT population in gestational age, birth weight, proportion of boys, singleton pregnancies, spontaneous onset of preterm labour and mode of delivery (table 1).

### DNA extraction and quality

DNA was extracted from a subset of blood samples from 521 full parent–infant trios who had successful linkage to infant clinical data. Of these, nine trios were excluded from genotyping due to inadequate DNA sample concentration (below 50 ng/µL) of one of the trio members. An additional two trios were excluded due to array capacity, resulting in 510 (97.9%) trios of this subset being successfully genotyped.

### Follow-up

Of the 883 infants whose parents consented to participate in the study, 796 (90.1%) agreed for their child's future health and educational data to be linked to the research data. Consent was also obtained from the parents of 794 (89.9%) for future direct contact.

### DISCUSSION

Large-scale prospective cohort studies are time-consuming and expensive to conduct. Cohorts of special patient or high-risk groups, such as preterm infants, require assembly from multiple centres across large geographical areas, which poses additional logistical challenge. In our pilot study, we overcame some of these barriers by optimising research operational efficiencies. Although other large-scale preterm neonatal cohorts with genetic data exist internationally,[13–17] none have collected parent DNA samples for trio-based genetic analyses. Parent blood samples for trio-based analysis are essential to identify likely pathogenic de novo variants. Our study approaches were purposefully designed to reduce research burden, respectful of the needs of preterm neonates and their families and sensitive to the culture within neonatal units. Within 1 year, we successfully established a large contemporaneous VPT neonatal cohort for genetic investigations.

### Choice of recruitment and data collection methods

The completeness and quality of data held in the NNRD has been validated for use in research.[18] The advantages

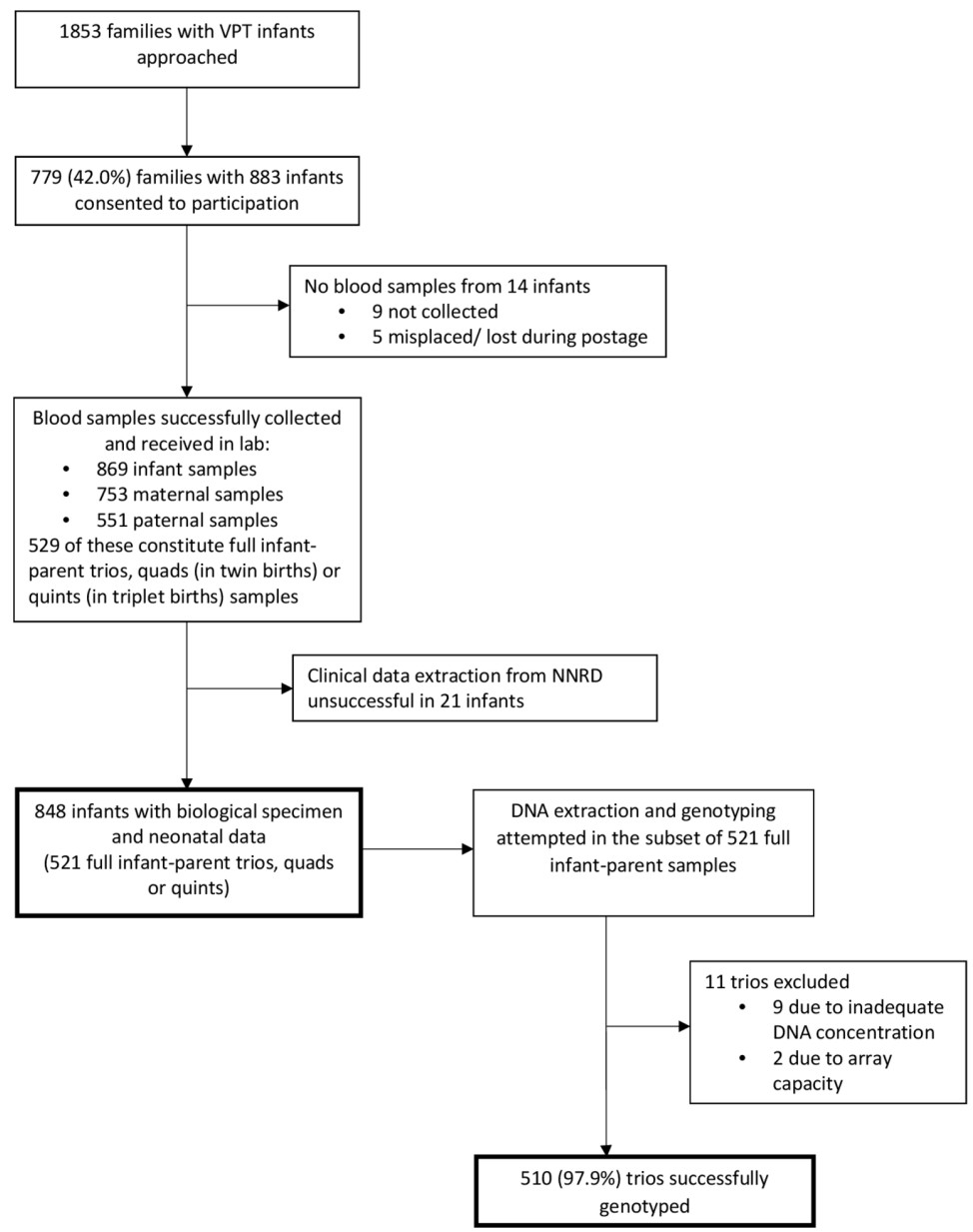

**Figure 1** Flowchart of recruitment and establishment of the cohort for the Study of Preterm Infants and Neurodevelopmental Genes, England, UK, 2017–2018. VPT, very preterm.

of leveraging routine electronic health records for research are well-recognised. These include less intrusive data collection, elimination of the need for duplicate data entry and reduced research burden and cost. The reduced demands allowed us to incorporate the study activities within the workload of existing clinical and research staff at the participating centres, thereby minimising research costs.

As no research-specific prospective data collection was required, our study design also removed the time pressure to recruit newborn infants immediately after birth, particularly as this may be a stressful and emotional time for parents. Time constraints on decision-making reduce willingness to participate.[19] Participants in our study were usually recruited a few weeks after delivery, when the

parents had become more familiar with the neonatal unit environment and the clinical staff.

Another strategy we adopted was to use opportunistic blood sampling for the infant. An aim of neonatal research is to minimise painful interventions and blood loss. Our approach removed the need for research-specific venepuncture and allowed sampling to be conducted flexibly.

### Limitations of study design

It was not possible to evaluate if our study design had a direct impact on recruitment rates as recruitment to neonatal studies varies widely depending on the nature of the study and the target population. Enrolment rates of more than 90% were achieved by population-based

**Table 1** Neonatal characteristics of SPRING participants and of all VPT infants born in the participating/study sites and in England during the recruitment period (May 2017– June 2018)

| | SPRING participants (n=848) | All eligible infants born in study sites (n=4491) | National cohort (n=8998) |
|---|---|---|---|
| Gestational age in weeks, median (IQR) | 28.9 (26–30) | 29 (27–30) | 29 (27–30) |
| Birth weight in grams, median (IQR) | 1120 (886–1420) | 1175 (882–1480) | 1170 (880–1470) |
| Boys, n (%) | 456 (53.8) | 2469 (55.0) | 4900 (54.5) |
| Singleton pregnancy, n (%) | 638 (75.2) | 3359 (74.8) | 6706 (74.5) |
| Spontaneous onset of labour, n (%) | 474 (55.9) | 2431 (54.1) | 4815 (53.5) |
| Caesarean section delivery, n (%) | 498 (58.7) | 2584 (57.5) | 5087 (56.5) |
| Maternal age at birth in years, median (IQR) | 31 (27–35) | Information not available | Information not available |

SPRING, study of preterm infants and neurodevelopmental genes; VPT, very preterm.

preterm birth cohorts using traditional study methods such as the EPIPAGE 2,[20] the EPICure 2[21] and the Express.[22] This suggests that parents of preterm infants are in general willing to participate in birth cohort studies. However, it is recognised that genetic research studies typically experience lower recruitment rates than other types of biomedical research.[23] Our enrolment rate (42.7% of parents approached) is comparable to that reported by the German Neonatal Network, which sought to collect DNA samples from very low birthweight (<1500 g) infants at 54 German NICUs and successfully enrolled 54% of eligible infants in the first 2 years.[24] As our study relied entirely on existing neonatal staff, it was possible that recruitment was reduced by competing workloads. Staffing availability and research capacity across the study centres were highly variable. This had a measureable impact on the implementation of the study with approximately one-third of participating neonatal units only initiating recruitment mid-way through the recruitment phase. Despite this, we managed to achieve our recruitment target of 500 families within 11 months, ahead of schedule. We did not include the recruitment of a 'term control group', which will be vital for direct case–control comparison, in this pilot study. The recruitment of well, term-born infants and their parents for genetic

analyses will undoubtedly lend new logistical and ethical challenges that need further exploration.

Although our cohort was representative of the national VPT population in neonatal characteristics, selection bias might have been introduced due to the lack of study information materials adapted for non-English speaking families and for those with low literacy. Ethnic and socioeconomic disparities in preterm birth rates are well-described. Future studies should consider how to better integrate study processes targeted at enabling participation of these groups.

Although the NNRD has been successfully utilised for clinical research in the UK, the utility of the database for the study of environmental influences and outcome phenotyping in the context of genetic investigations has yet to be demonstrated, and will need to be examined in future studies. We had not allowed for the collection of different biological tissues at different time points to enable epigenetic analyses, as this would have added onto the research burden of the clinical teams and the families participating. The feasibility and acceptability of using archived or surplus biological specimens collected during routine clinical care, for example archived newborn blood spot, for this purpose could be further explored.

### Acceptability of research by data linkage

The increasing availability of electronic health records and administrative data provides opportunity for longitudinal cohort follow-up by data linkage. Traditional methods used in neonatal follow-up studies can be time-consuming, expensive and burdensome and are particularly problematic for large sample sizes. In addition, follow-up studies are often compromised by selection bias due to attrition. We found a high level of willingness among parents for their child's routine data to be used in neonatal follow-up research. This corroborates the findings from a systematic review that reported widespread public support for data sharing and linkage for research purposes.[25] In our study, we sought broad consent without specifying the time and type of data that will be accessed. However, Audrey *et al* revealed the complexity of issues surrounding informed consent for data linkage in a large qualitative study of young people's views.[26] Four hypothetical scenarios were presented to 55 participants to prompt considerations regarding consent requirements. The authors concluded that perceptions were diverse, and sometimes inconsistent. In general, participant views on data linkage were influenced by the social sensitivity of the research question and the likely health benefits in the public interest. Future research in data linkage will benefit from a robust and consistent governance framework that is acceptable to the public.

### CONCLUSIONS

We demonstrate the feasibility of a streamlined study approach to achieve a large prospective VPT cohort

for clinical and genetic studies and the acceptability of follow-up through data linkage and recall. Our initial cohort is an important resource for studies of genetic risks and preterm birth, and longitudinal health and educational outcomes; our planned larger cohort will amplify this resource. The strategies described provide a template for future cost-effective and efficient cohort development.

**Acknowledgements** We are grateful to the participating families and staff at all the neonatal units. We thank R Colquhoun, Research Manager and K Ougham, Data Analyst at the Neonatal Data Analysis Unit, Imperial College London for technical and administrative assistance as well as L Bates, J Morgan, N Vinh, A Evans, L Tram, S Jaques and S Lewis at the MRC Centre for Neuropsychiatric Genetics and Genomics, Cardiff University School of Medicine, United Kingdom for laboratory and technical assistance.

**Contributors** NM and AT conceptualised this study; NM, AT and MCO'D designed the study; AT, MCO'D, NM and HSW obtained funding. HSW and LH acquired and analysed the data. HSW wrote the first draft of the manuscript. All authors contributed to the editing of the manuscript and have approved the final manuscript.

**Funding** This study was funded by the Medical Research Council, UK (reference MR/N025288/1).

**Competing interests** NM reports grants from the Medical Research Council, National Institute of Health Research, Department of Health, Westminster Research Fund, Prolacta LIfe Sciences, Chiesi Pharmaceuticals, grants from Shire Pharmaceuticals, HCA International, Healthcare Quality Improvement Partnership, March of Dimes. NM is director of the Neonatal Data Analysis Unit, chief investigator for the National Neonatal Research Database, and coinvestigator for the Department of Health Funded Maternal and Neonatal Health and Care Policy Research Unit at the University of Oxford. In the last three years, NM has served on the Nestle Scientific Advisory Board and the Boards of Trustees of the Royal College of Paediatrics and Child Health, David Harvey Trust, Medical Women's Federation, Medact, Action Cerebral Palsy and TheirWorld.

**Patient consent for publication** Not required.

**Ethics approval** This study was reviewed and approved by the Health Research Authority and the National Health Service Research Ethics Committee (reference 16/WA/0324).

**Provenance and peer review** Not commissioned; externally peer reviewed.

**Data availability statement** Data sharing not applicable as no datasets generated and/or analysed for this study.

**ORCID iD**
Hilary S Wong http://orcid.org/0000-0003-4597-1794

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
