## [Reviewer comments · BMJ Paediatrics Open]

ARTICLE DETAILS

TITLE (PROVISIONAL)	Pilot study to establish a prospective neonatal cohort: Study of Preterm Infants and Neurodevelopmental Genes (SPRING)
AUTHORS	Wong, Hilary; Hopkins, Lucinda; O'Donovan, Michael; Thapar, Anita; Modi, Neena

VERSION 1 – REVIEW

REVIEWER	Reviewer name: Sharon Casavant Institution and Country: University of Connecticut United States Competing interests: None
REVIEW RETURNED	11-Feb-2020

GENERAL COMMENTS	Thank you for the opportunity to review this protocol paper. The authors are truly fortunate to have access to such a database. My critiques are minor: 1) In Background, the first sentence has an inappropriate reference, these statistics actually come from the World Health Organization. Using a systematic review is not a primary source and should be avoided.2) In the second paragraph of the Background, the sentence that begins "Whilst genomic studies..." is awkward and lengthy. Consider revising it to two sentences.3) Under Methods, Study sites and recruitment, please describe the differences between a NICU and an LNU in the UK.4) Under Biological specimen collection, you don't describe how you checked the quality (you have that as a separate piece under results). I think if you're going to define how you quantified the DNA, you need to define how you did quality control.5) Under Clinical Data Collection, please describe the point-of-care electronic data (daily records of interventions and treatments that follows infants to two years of age)6) In Consent for follow-up: remove the word "question" and replace with "statement".7) In the same section the statement is awkward, you need to remove the word "to" so that the statement reads "I give permission for the research team to link the information my baby as part of the study records..."8) Under DNA extraction and quality, please describe what technique you used to determine quality. PCR? Overall, this is a great protocol paper for researchers who have access to databases such as yours.
---

REVIEWER	Reviewer name: karel allegaert Institution and Country: KU Leuven, Belgium and Erasmus MC, Rotterdam, the Netherlands Competing interests: i have co-authorships with the final author of this paper
REVIEW RETURNED	12-Mar-2020

GENERAL COMMENTS	I highly value the 'pragmatic' approach described in the current pilot study, as the existing national neonatal research database (NNRD), so this is a valuable paper. This approach likely does not only reduce the burden for researchers, but also the threshold for parents. The issue is perhaps on the 'consent for re-contacting the family' so that 'opting out' may occur more common at a second time, or is this also only based on the medical files already available ? The authors describe indeed the feasibility to collect these data, but it is not yet proven that the granularity of the data is sufficiently well to have results beyond the genetic profiles. Do I understand well that issues like disease severity, pharmacotherapy or nutritional strategies can potential be linked, but are not yet part of the current study protocol (how eg to handle maternal therapy, prenatal steroids, other environmental issues, like eg the growing concern on acetaminophen ? where are the 'outcome variables ? and related to the reference made to the German BPD study, why 'only' neuropsychology/autism related issues. How does this compare to the 'historical' twin type of approaches to explore genetics to environment ? Is there already a preplanned framework how to compare data in preterm to 'similar' term cohorts? No epigenetic approaches included or considered? In essence, this paper illustrates indeed that trio genetics collected is feasible and linked to clinical registration, but still has to proof that this will result in sufficiently robust data.
--

REVIEWER	Reviewer name: Wolfgang Göpel Institution and Country: University of Lübeck, UKSH Germany Competing interests: none
REVIEW RETURNED	16-Mar-2020

GENERAL COMMENTS	1. The title of the study refers to neurodevelopmental genes as the main target for future analyses. In the discussion section, (page 12, line 8) they state that they "sought broad consent without specifying the time and type of data that will be accesses". I was wondering if the authors are favouring a more limited approach of data collection and analyses (which is implied by the title of the article) or a broad consent including other outcomes than neurodevelopment. 2. The study was funded by the UK Medical Research Council but other large scale genetic data are owned companies like 23andMe and deCODE. Was data ownership and the source of funding an issue in discussion with parents and Bliss?
---

VERSION 1 – AUTHOR RESPONSE

Reviewer: 1

Thank you for the opportunity to review this protocol paper. The authors are truly fortunate to have access to such a database. My critiques are minor:

1) In Background, the first sentence has an inappropriate reference, these statistics actually come from the World Health Organization. Using a systematic review is not a primary source and should be avoided.

Response: This paper referred to in the reference was co-authored by WHO staff as the key report that details the primary data sources and the modelling methods utilised by WHO to provide the global estimates of preterm birth. This was referred to in the WHO bulletin: <https://www.who.int/news-room/detail/17-11-2018-new-global-estimates-on-preterm-birth-published>. As such, we have kept this reference in the paper.

2) In the second paragraph of the Background, the sentence that begins "Whilst genomic studies..." is awkward and lengthy. Consider revising it to two sentences.

Response: We have revised this sentence into two sentences

3) Under Methods, Study sites and recruitment, please describe the differences between a NICU and an LNU in the UK.

Response: We included the following description: "In the UK, neonatal care is provided within managed clinical networks where LNU provide the full range of care, including short periods of intensive care, for babies delivered in the nominal catchment area of the hospital [11]. Most babies born over 27 weeks gestation will usually receive their full care in their LNU. Babies who are more preterm or those who require complex or longer-term intensive care are transferred to a NICU, where tertiary specialist care is provided."

4) Under Biological specimen collection, you don't describe how you checked the quality (you have that as a separate piece under results). I think if you're going to define how you quantified the DNA, you need to define how you did quality control.

Response: We have added the following sentences to the section: "Samples with concentrations below 50ng/ul were excluded from genotyping. DNA was considered of adequate quality on successful completion of array based genotyping."

5) Under Clinical Data Collection, please describe the point-of-care electronic data (daily records of interventions and treatments that follows infants to two years of age)

Response: We have included the following sentence: "Data captured on the NNRD includes demographic details, daily records of intervention and treatments through the neonatal inpatient stay, diagnoses and follow-up health status at age two years."

6) In Consent for follow-up: remove the word "question" and replace with "statement".

Response: We have made the change as suggested.

7) In the same section the statement is awkward, you need to remove the word "to" so that the statement reads "I give permission for the research team to link the information my baby as part of the study records..."

Response: We have altered the sentence to "I give permission for the research team to link the study information on my baby to records that the NHS and other public organisations hold". We are uncertain which word "to" that the reviewer suggests removing. However, we have kept the words necessary to convey the essence of the sentence.

8) Under DNA extraction and quality, please describe what technique you used to determine quality. PCR?

Response: As in point 4, we clarified in the methods section that the DNA sample concentrations below 50ng/ul were excluded and DNA were considered of adequate quality on successful completion of array based genotyping.

Overall, this is a great protocol paper for researchers who have access to databases such as yours.

Reviewer: 2

I highly value the 'pragmatic' approach described in the current pilot study, as the existing national neonatal research database (NNRD), so this is a valuable paper. This approach likely does not only reduce the burden for researchers, but also the threshold for parents. The issue is perhaps on the 'consent for re-contacting the family' so that 'opting out' may occur more common at a second time, or is this also only based on the medical files already available ?

Response: We designed the study such that there are two approaches for follow-up. The first approach is to utilise routine data linkage for future outcome ascertainment. The second approach and the purpose for gaining consent to 're-contact the family' is to facilitate future data collection through face-to-face or questionnaire-based assessments. Parents were made aware of their right to withdraw their consent at any point in the study and in the future by contacting the research team. We have explained further this under the section Methods/ Consent for follow up.

The authors describe indeed the feasibility to collect these data, but it is not yet proven that the granularity of the data is sufficiently well to have results beyond the genetic profiles.

Do I understand well that issues like disease severity, pharmacotherapy or nutritional strategies can potential be linked, but are not yet part of the current study protocol (how eg to handle maternal therapy, prenatal steroids, other environmental issues, like eg the growing concern on acetaminophen ? where are the 'outcome variables ? and related to the reference made to the German BPD study, why 'only' neuropsychology/autism related issues.

Response: The study was limited to neuropsychiatric disorders as it acts as an exemplar study to pilot the study processes and test the feasibility of establishing a neonatal cohort with genetic data for future clinical and genetic investigations. The reviewer is correct that data in the NNRD, which includes disease severity, pharmacotherapy and nutritional strategies can potentially be linked. The NNRD has been validated with several examples of these data used in research studies in the UK. However, there is no power within this current pilot study to examine the interaction of genetics and environmental variables.

How does this compare to the 'historical' twin type of approaches to explore genetics to environment ?

Response: As mentioned above, the current pilot study is not powered to examine the interaction of genetics and environmental variables and the aim of this cohort is different to twin approaches. Twin studies infer genetic contributions by examining monozygotic and dizygotic concordance and discordance. This study set out to assess the feasibility of collecting DNA from infants and parents for the ultimate purpose of directly examining DNA variation (rare and common variation).

Is there already a preplanned framework how to compare data in preterm to 'similar' term cohorts?

Response: We did not recruit 'control' term infants in this pilot study. When we conducted preliminary investigation to explore the hypothesis that the very preterm population is enriched for rare copy number variants (CNVs), we compared the rate of CNVs in this VPT cohort to published rates in control populations of parent-proband trios as well as UK Biobank participants (see publication <https://jmg.bmj.com/content/early/2020/02/12/jmedgenet-2019-106619>). We intend to include term infants in future recruitments. We have included the lack of term-born control as a limitation in the discussion section.

No epigenetic approaches included or considered?

Response: Information on epigenetic influences were not collected within the scope of this study. Epigenetic analyses are tissue-specific and the multiple collection of different biological specimens at different time-point will increase the research burden and thereby reducing the intended efficiencies of this pilot study. We have also included the lack of epigenetic data in the discussion section.

In essence, this paper illustrates indeed that trio genetics collected is feasible and linked to clinical registration, but still has to proof that this will result in sufficiently robust data.

Response: We thank the reviewer for these comments and have added this point in the discussion.

Reviewer: 3

1. The title of the study refers to neurodevelopmental genes as the main target for future analyses. In the discussion section, (page 12, line 8) they state that they "sought broad consent without specifying the time and type of data that will be accesses". I was wondering if the authors are favouring a more limited approach of data collection and analyses (which is implied by the title of the article) or a broad consent including other outcomes than neurodevelopment.

Response: This pilot study was set up to facilitate the preliminary analyses on the hypothesis that the very preterm population is enriched for copy number variants (CNVs) because those born preterm show an elevated rate of neurodevelopmental disorders like autism. The hypothesis for this feasibility project was that CNVs known to contribute to neurodevelopmental disorders also are associated with preterm birth, hence the title of the study (The genetic findings have been published elsewhere (<https://jmg.bmj.com/content/early/2020/02/12/jmedgenet-2019-106619>)). Participants were initially recruited to contribute blood/ DNA samples to study this hypothesis. However, when we sought further consent from parents for data linkage, this was a 'broad consent' to allow further study of the relationship between genetics and other neonatal factors with future health and development of children born preterm.

2. The study was funded by the UK Medical Research Council but other large scale genetic data are owned companies like 23andMe and deCODE. Was data ownership and the source of funding an issue in discussion with parents and Bliss?

Response: The funding source of the study and the indefinite storage of data and genetic material (as part of a research tissue bank) was fully stated in the parent information sheets provided to families prior to enrolment in the study.